# Unveiling hub genes and biological pathways: A bioinformatics analysis of Trauma-Induced Coagulopathy (TIC)

Lingang Zhang[1,*,◦], Bo Li[2,◦], Jing Liu[3], Yan feng Bian[4], Guo xing Lin[5], Ying Zhou[1]

**1** Emergency Department, Yuncheng Central Hospital affiliated to Shanxi Medical University,Yuncheng, Shanxi, China, **2** Reproductive Medicine Department, Yuncheng Central Hospital affiliated to Shanxi Medical University, Yuncheng, Shanxi, China, **3** Pathology Department, Yuncheng Central Hospital affiliated to Shanxi Medical University,Yuncheng, Shanxi, China, **4** Emergency sungery, Shanxi Bethune Hospital, Shanxi Academy of Medical Sciences,Tongji Shanxi Hospital,Third Hospital of Shanxi Medical University, China, **5** Emergency Department, Hebei province Xingtai Third People's Hospital, Xingtai, China

◦ These authors contributed equally to this work.

* zhanglinggang@163.com

## Abstract

### Background

Trauma-Induced Coagulopathy is a severe condition that rapidly manifests following traumatic injury and is characterized by shock, hypoperfusion, and vascular damage. This study employed bioinformatics methods to identify crucial hub genes and pathways associated with TIC.

### Methods

Microarray datasets (accession number GSE223245) were obtained from the Gene Expression Omnibus (GEO) database. The data were subjected analyses to identify the Differentially Expressed Genes (DEGs), which were further subjected to GO and KEGG pathway analyses. Subsequently, a Protein-Protein Interaction (PPI) network was constructed and hub DEGs closely linked to TIC were identified using CytoHubba, MCODE, and CTD scores. The diagnostic value of these hub genes was evaluated using Receiver Operating Characteristic (ROC) analysis.

### Results

Among the analyzed genes, 269 were identified as DEGs, comprising 103 upregulated and 739 downregulated genes. Notably, several significant hub genes were associated with the development of TIC, as revealed by bioinformatic analyses.

**Data availability statement:** The dataset used in our study is publicly available in the GEO database under the accession number GSE223245. The direct link to access the dataset is: https://www.ncbi.nlm.nih.gov/geo/query/acc.cgi?acc=GSE223245

**Funding:** 2023 Yuncheng City Basic Research Program (Free Exploration Category) Projects Item No: YCKJ-2023052. The funders had no role in study design, data collection and analysis, decision to publish, or preparation of the manuscript.

**Competing interests:** The authors have declared that no competing interests exist.

## Conclusions

This study highlights the critical impact of newly discovered genes on the development and progression of TIC. Further validation through experimental research and clinical trials is required to confirm these findings.

---

## 1. Introduction

Uncontrolled hemorrhage is a significant preventable factor that contributes to mortality in patients with traumatic injuries. Additionally, among individuals under the age of 50 years, injury is the second most significant cause of mortality, closely following infectious diseases [1]. Impaired coagulation following sudden death due to injury has been recognized and recorded for centuries [2]. TIC refers to the abnormal coagulation processes that occur as a result of trauma. Severe trauma can result in the development of TIC through various mechanisms, including activation of protein C, disruption of the endothelial glycocalyx, consumption of fibrinogen, and platelet dysfunction. The ultimate objective of personalized medicine for patients at risk of TIC is to ensure the delivery of the most suitable products to each individual patient at the right time. Recent studies have highlighted specific molecular factors that play critical roles in the pathogenesis of TIC. Tissue Factor (TF) plays a pivotal role in TIC. Upon endothelial injury, TF is exposed and binds with factor VIIa, activating the extrinsic coagulation pathway, leading to thrombin generation and fibrin formation [1]. Platelet Factor 4 (PF4), released during platelet activation, is closely associated with platelet dysfunction and hypercoagulability in TIC. Studies have demonstrated its role in promoting a procoagulant state and contributing to the progression of TIC [1]. However, despite significant research efforts, our current understanding of the pathophysiology of TIC remains incomplete. This incompleteness is further compounded by limitations in diagnostic testing, which contributes to the imprecision of current clinical decisions.

The rapid development of innovative technologies, including next-generation sequencing (NGS), has significantly accelerated the exploration of diagnostic and therapeutic biomarkers for TIC.

Bioinformatics analysis plays a crucial role in uncovering novel clues and essential data for the identification of reliable and functional differentially expressed genes (DEGs) and non-coding transcripts [3]. Furthermore, integrated studies that combine data from various medical sources not only save resources, but also provide valuable evidence for mapping the molecular pathogenesis networks of diseases.

In this study, we assessed the gene expression profile for traumatic coagulopathy from the GEO database. The GEO database is an open-access resource that offers comprehensive genetic information, making it a valuable tool for bioinformatic analysis and identification of new disease targets [4]. Using bioinformatics methods, we successfully identified differentially expressed genes (DEGs). DEGs were subsequently subjected to analysis using protein–protein interactions (PPI), Gene Ontology (GO) and Kyoto Encyclopedia of Genes and Genomes (KEGG) pathways.

Furthermore, the study analyzed the pathways associated with TIC, as well as the interactions between DEGs and pathways. Subsequently, the hub DEGs were identified using CYTOHubba, MCODE, and CTD scores. CYTOHubba is a Cytoscape plugin used to identify key hub genes within molecular networks. MCODE helps detect densely connected modules in large-scale networks, while CTD is a database that provides insights into gene-disease and chemical-gene interactions. Lastly, Receiver Operating Characteristic (ROC) analysis was conducted to evaluate the diagnostic value of the identified hub genes. The ROC curve is a graphical tool used to evaluate the performance of binary classification models by plotting the True Positive Rate (TPR) against the False Positive Rate (FPR) at various thresholds. The Area Under the Curve (AUC) quantifies the overall ability of the model to distinguish between classes, with higher AUC values indicating better performance. As a result, the identified hub genes have the potential to become a novel area of research focus. The elucidated molecular mechanisms and signaling pathways may provide valuable insights into understanding TIC.

## 2. Methods

### 2.1 Microarray data retrieval

Coagulopathy datasets were acquired from the National Center for Biotechnology Information (NCBI) GEO (http://www.ncbi.nlm.nih.gov/geo) [5] public repository. Lastly, we obtained GSE223245 from the NCBI GEO. The GSE223245 dataset was generated using the GPL33038 platform, which includes ceRNA chipset samples from Homo sapiens. This dataset comprised 12 patients with traumatic brain injury (TBI) and 4 healthy controls, with peripheral blood mononuclear cells (PBMC) collected for analysis.

### 2.2 Data Processing and Differentially Expressed Genes Identification

Microarray data were accessed from GEO using the R package "GEOquery." Differentially expressed genes (DEGs) were obtained from the microarray data using the R package "limma." All identified differentially expressed genes (DEGs) met the criteria of p-value < 0.05 and log2 (fold-change) ≥1. The resulting differentially expressed genes (DEGs) were visualized using a Volcano Plot created using the R packages "ggplot2"[6] and "dplyr." Additionally, a Heatmap was generated using the R package "pheatmap" to further visualize DEGs.

### 2.3 Functional enrichment analysis

Gene Set Enrichment Analysis (GSEA) [7] was performed using the R package "clusterProfiler" [8]. The process of conducting Gene Ontology (GO) and Kyoto Encyclopedia of Genes and Genomes (KEGG) pathway enrichment analyses of Differentially Expressed Genes (DEGs) were performed using the R package "clusterProfiler" [8]. The results were visualized using the R packages "ggplot2" [6].

### 2.4 Analysis of PPI and identifcation of Hub genes

Protein-protein interaction network construction and module analysis were performed following established protocols, as described in previous studies [9]. The overlapping DEGs were subjected to protein-protein interaction (PPI) analysis using the STRING database (https://string-db.org/) [10]. Network visualization of the resulting interactions was achieved using Cytoscape version 3.8.2 [11]. Hub genes were identified using CytoHubba and MCODE plugins, which were implemented in Cytoscape version 3.8.2.

### 2.5 Expression and ROC analysis

Based on their centrality values, the top ten genes in the protein-protein interaction (PPI) network were identified as critical genes. Receiver operating characteristic (ROC) curves were generated using the pROC R package to analyze the performance of the classification model, and ROC curves were used to assess the predictive capability of the identified biomarkers.

## 2.6 Ethical approval

The data for this study were obtained from the public GEO database, and ethical committee approval was not required.

## 3. Results

### 3.1 Identification of differentially expressed genes

A flowchart depicting the overall data-screening strategy is shown in Fig 1. In the coagulopathy dataset GSE223245, 823 DEGs were filtered when we compared the 12 coagulopathy samples with 4 healthy controls. In the GSE223245 dataset, the differential analysis revealed a total of 269 differentially expressed genes (DEGs), with 103 genes up-regulated and 166 genes down-regulated in coagulopathy samples compared to healthy samples.The DEGs were visualized through Volcano Plots, Heatmaps, PCA, and Boxplots, providing comprehensive insights into their expression patterns. (Fig 2A-D). We observed a correlation between the DEGs (Fig 3a, b)

### 3.2 GO and KEGG enrichment pathway analysis

To explore the functional aspects of the DEGs more comprehensively, gene symbols were analyzed using the latest versions of the GO and KEGG pathway databases. This analysis aimed to ascertain the potential functions associated with DEGs.

The results of the KEGG analysis demonstrated significant enrichment primarily in some pathways: tryptophan metabolism, autoimmune thyroid disease, steroid hormone biosynthesis, inflammatory bowel disease, natural killer cell-mediated cytotoxicity, viral protein interaction with cytoking and cytoking receptors, Pantothenate and CoA biosynthesis, and the NOD-like receptor signaling pathway (Fig 4a). GO analysis indicated that the DEGs were primarily enriched in

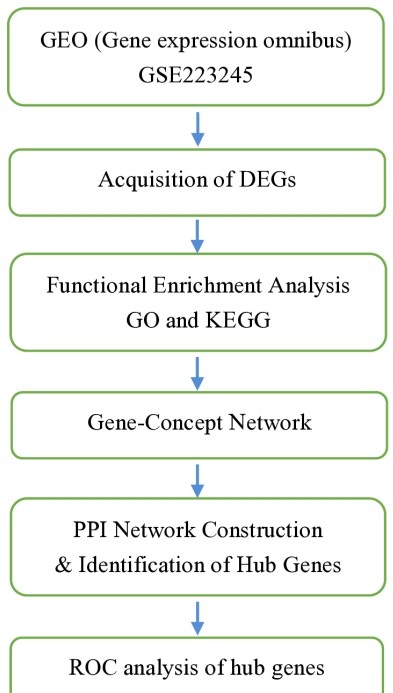

**Fig 1. The multistep screening strategy for bioinformatics data is presented in the flowchart below.**

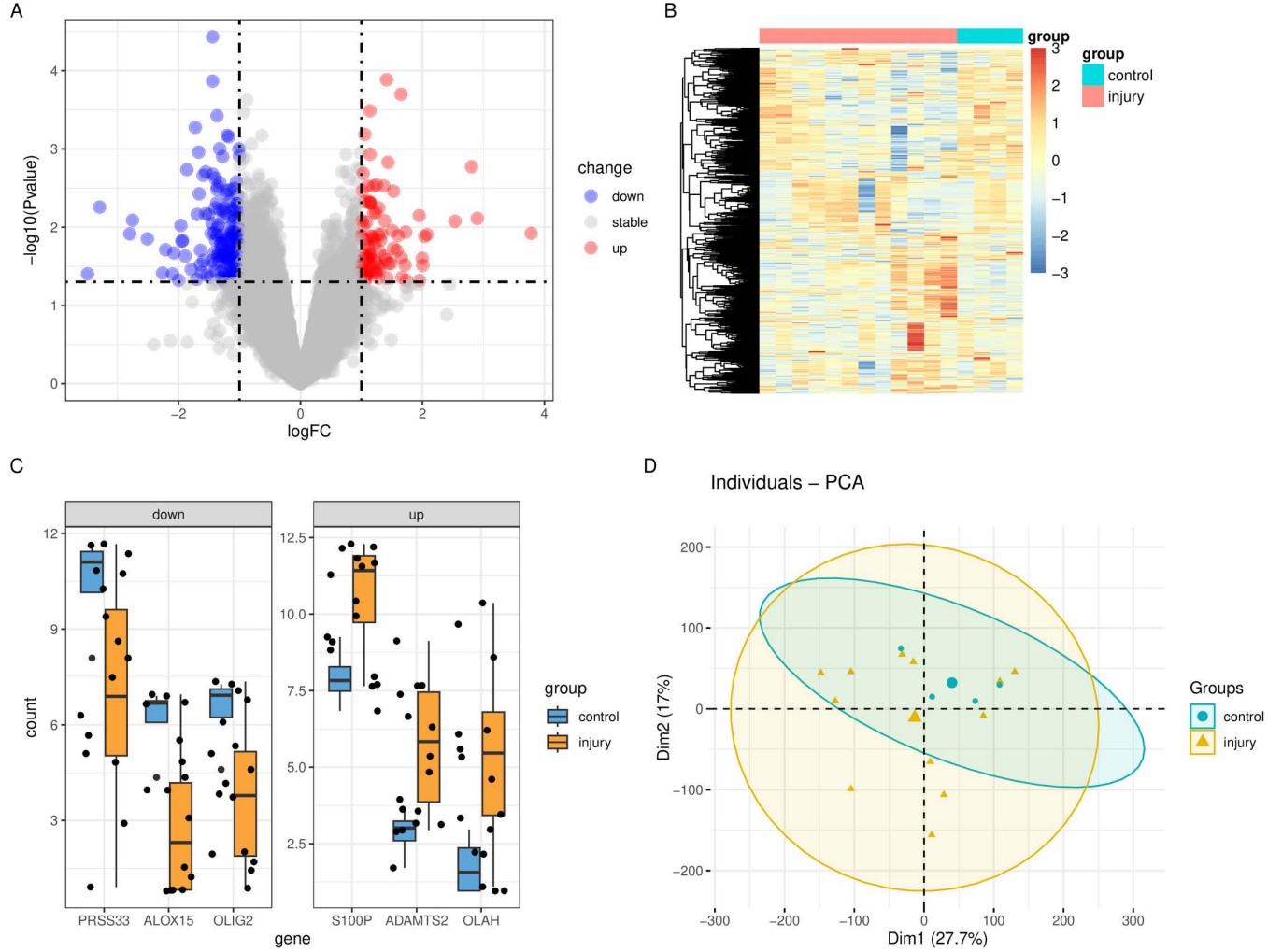

**Fig 2. DEG in traumatic coagulopathy. A** Volcano plot of DEGs in GSE223245; **B** Clustered heatmap of DEGs in GSE223245; **C** The box plot show-
cases the expression levels of the first three up-regulated genes and the last three down-regulated genes; **D** the PCA score plots show a comparison
between the coagulopathy group and the healthy group in the datasets.

some categories: regulation of immune effector process, regulation of response to biotic stimulus, regulation of cell killing, regulation of innate immune response, and defense response to virus (Fig 4b-c). These pathways play a significant role in the occurrence and progression of traumatic coagulopathy.

A correlation exists between the primary pathways and genes (Fig 5a-b, Fig 6b). These pathways primarily exhibit a high concentration in defense responses to viruses, regulation of cell killing, regulation of immune effector processes, regulation of innate immune responses, and regulation of responses to biotic stimuli. The genes intricately linked to these pathways are *CXCL6, CD160, KLRC4, DDX60, IFIT1, RSAD2, CFH, VSIG4.* It is possible to gain an understanding of the interrelationships between these pathways (Fig 6a). The pathways that exhibit the highest concentration are predominantly those regulating cell killing, regulation of immune effector processes, regulation of innate immune response, and regulation of response to biotic stimulus.

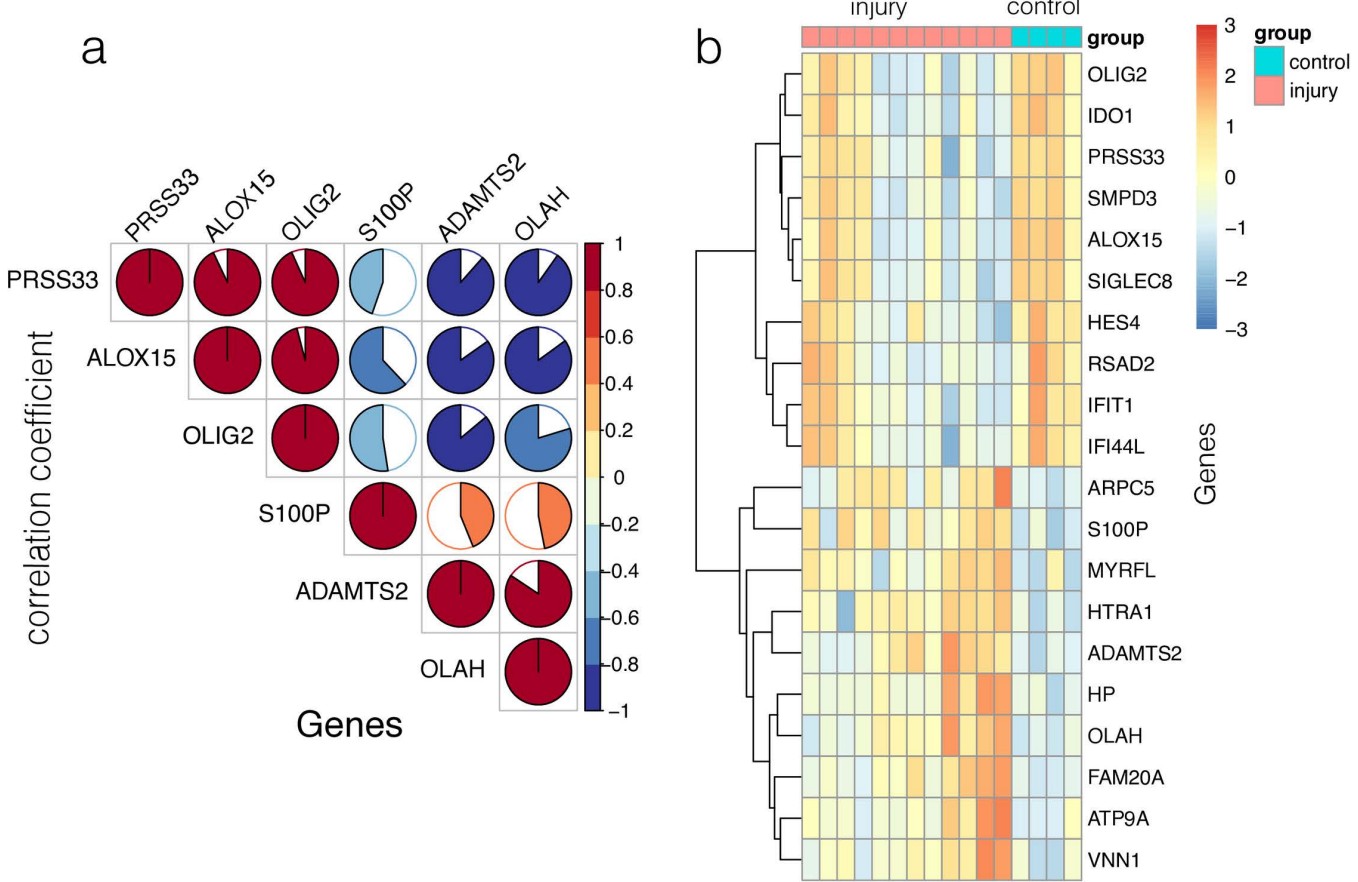

**Fig 3.** (a) The corrplot showcases the expression levels of the first three up-regulated genes and the last three down-regulated genes. (b) The Clustered heatmap showcases the expression levels of the first ten up-regulated genes and the last ten down-regulated genes.

### 3.3 Exploring PPI networks and identifying hub genes: uncovering key interactions in DEGs

A protein-protein interaction (PPI) network was constructed using the PPI pairs from the STRING database, representing the interactions among proteins encoded by the DEGs. The PPI network was visualized using Cytoscape, allowing for a comprehensive analysis of protein interactions (Fig 7a). Significant modules (gene clusters) were identified using the MCODE plugin, which facilitates the detection of densely connected regions within the PPI network.

The PPI network was analyzed using the MCC algorithm from the CytoHubba plugin, resulting in the identification of 10 hub genes as the top candidates. *OAS2, OAS3, IFIT2, IFIT1, IFIT3, HERC5, IFI44, IFI44L, RSAD2,* and *DDX60* (Fig 7b).

### 3.4 Assessing the diagnostic value of hub genes

To validate the diagnostic value of the 10 hub genes obtained from the previous analysis, ROC curves were constructed and the corresponding area under the curve (AUC) was calculated for gene expression levels in the traumatic coagulopathy datasets (Fig 8). The AUC for *OAS2, OAS3, IFIT2, IFIT1, IFIT3, HERC5, IFI44, IFI44L, RSAD2, DDX60* were 0.854, 0.833, 0.854, 0.854, 0.854, 0.812, 0.833, 0.833, 0.833, 0.875.

This Table 1 summarizes the 10 key hub genes identified in the study and their proposed roles in the pathogenesis of TIC.

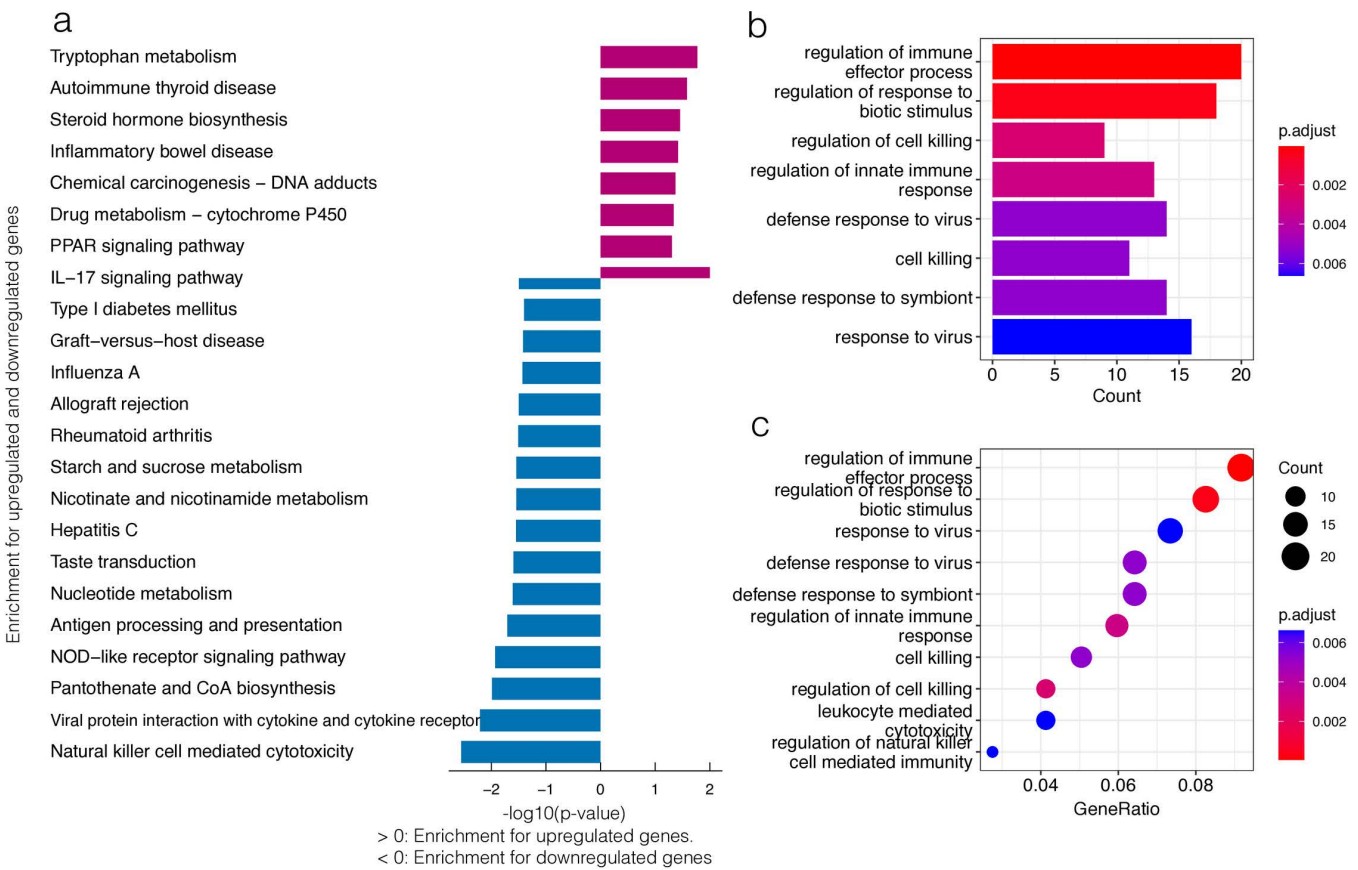

**Fig 4. Enrichment analysis of the DEGs.** (a) Outcomes of the KEGG enrichment analysis for the DEGs. (b and c) Outcomes of the GO functional analysis demonstrating the significantly enriched terms for the DEGs.

## 4 . Discussion

In the global context, injuries hold the position of being the fourth leading cause of mortality [1]. In civilian [12] and military [13] settings, early preventable deaths after injury are mainly caused by uncontrolled hemorrhage [12–17], whereas later preventable deaths are typically attributed to hypercoagulability [18]. After experiencing massive trauma with the presence of shock, hypoperfusion, and vascular damage, Trauma-Induced Coagulopathy (TIC) develops rapidly. This condition impairs the ability of the body to form blood clots and can lead to increased bleeding risk [19]. A comprehensive understanding of TIC pathophysiology is indispensable for lowering trauma-related mortality rates [20]. The mechanisms underlying TIC involve the activation of protein C, disruption of the endothelial glycocalyx, decreased fibrinogen levels, and impaired platelet function [19]. Nevertheless, the pathogenesis of TIC remains elusive and there is a scarcity of effective therapeutic strategies to address this condition. In this context, it is imperative to enhance our understanding of TIC pathogenesis and actively seek potential therapeutic targets. Using various bioinformatic methods, the current study successfully retrieved DEGs from TIC-related microarray datasets sourced from the GEO database. Furthermore, the study encompassed GO and KEGG pathway enrichment analyses. Subsequently, a PPI network was assembled to identify the top 10 hub genes among the DEGs. Ten hub genes (*OAS2, OAS3, IFIT2, IFIT1, IFIT3, HERC5, IFI44, IFI44L, RSAD2*, and *DDX60*) were selected to validate their diagnostic value in patients with TIC (P<0.05). These genes possess significant potential for predicting the risk of TIC, making them crucial candidates for further investigation.

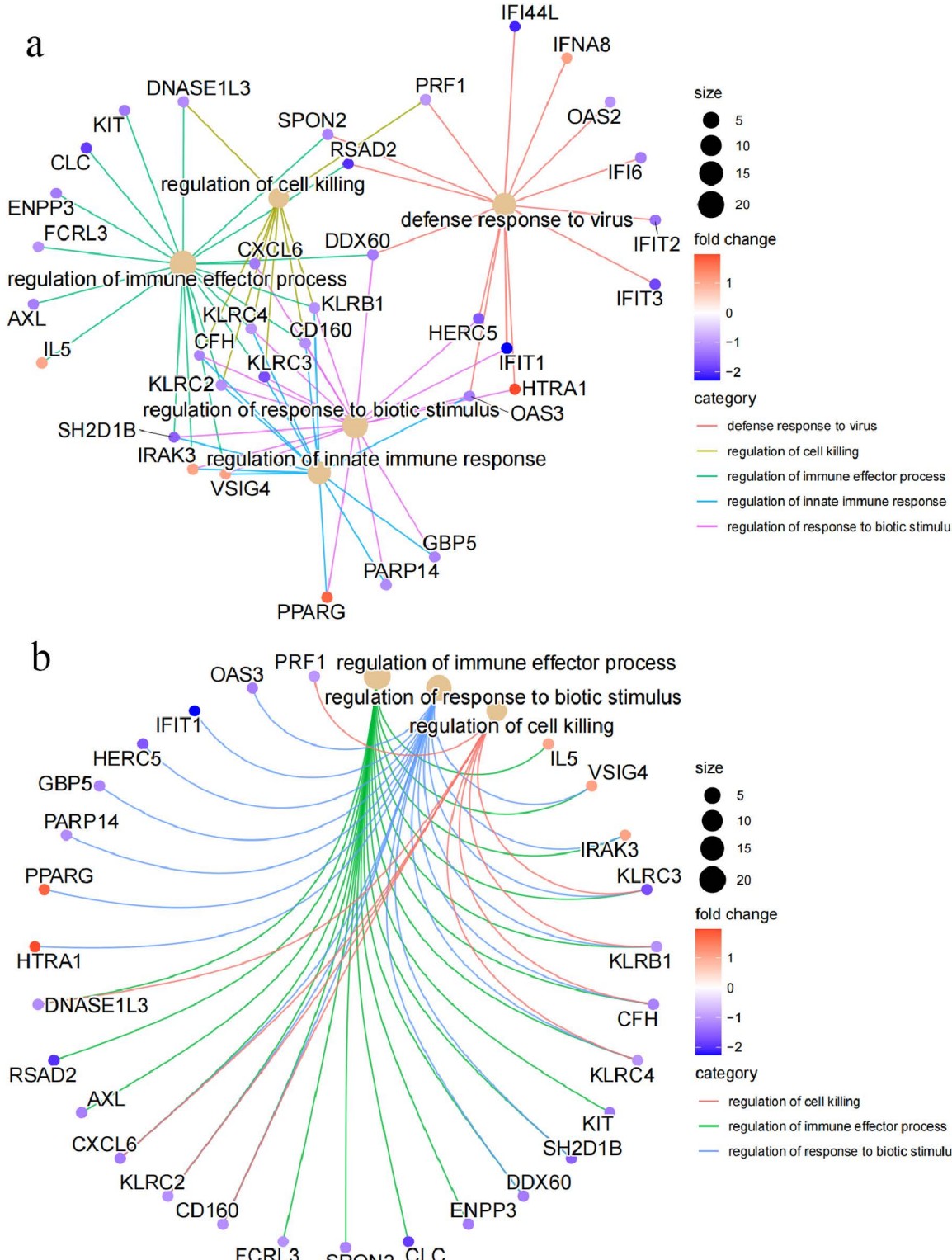

**Fig 5. The relationship between the main pathways and DEGs.**

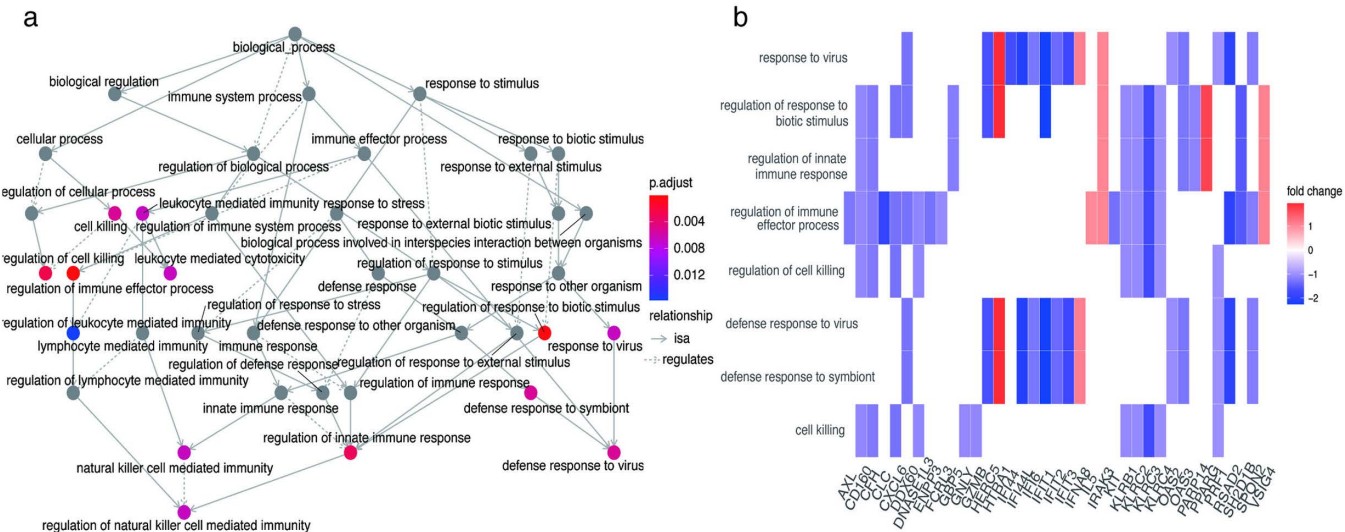

**Fig 6. There is a correlation between different pathways** (a). The Clustered heatmap showcases the relationship between the main pathways and DEGs (b).

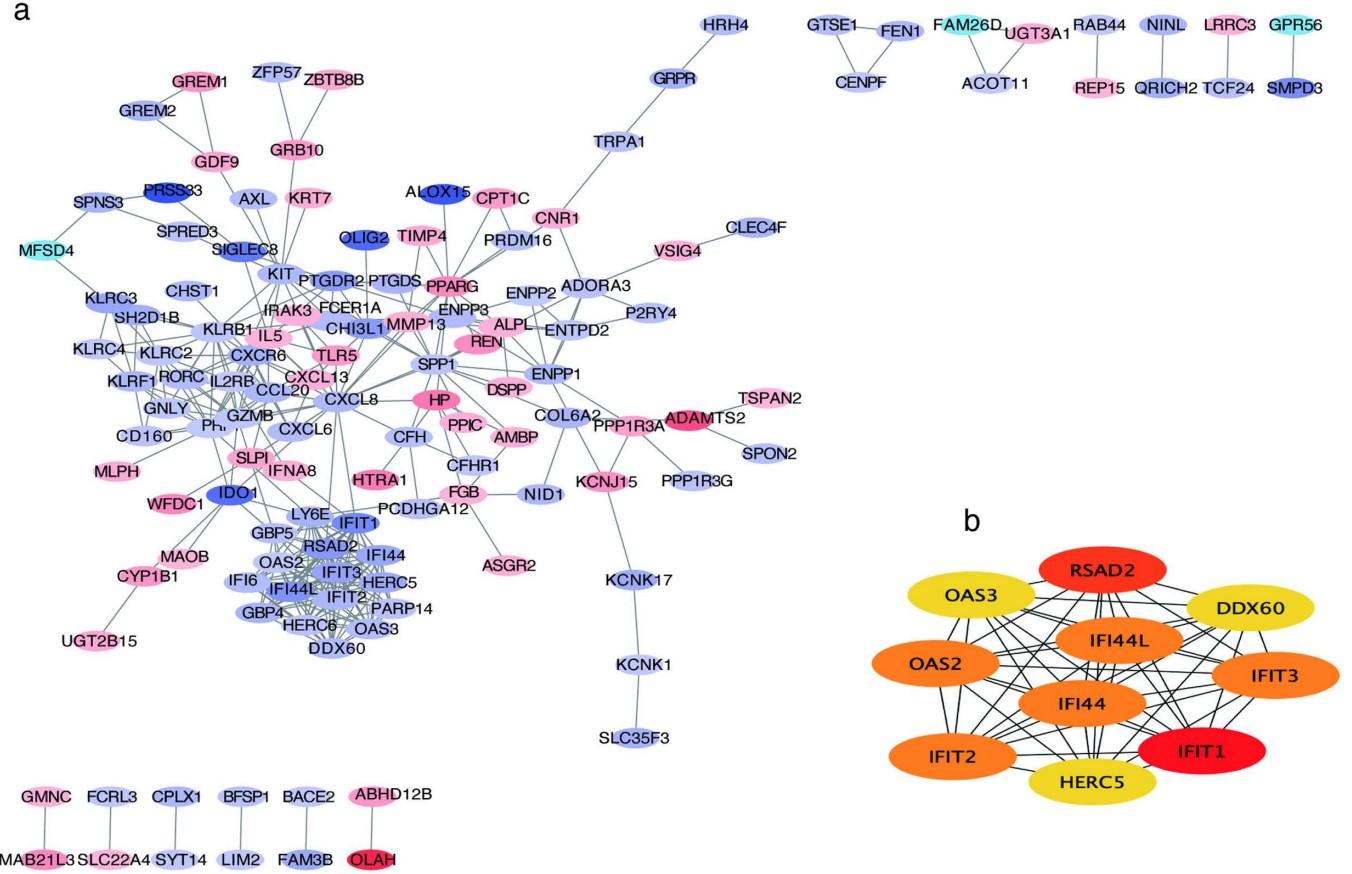

**Fig 7. PPI network construction (a); Top 10 hub genes explored by CytoHubba. (b).**

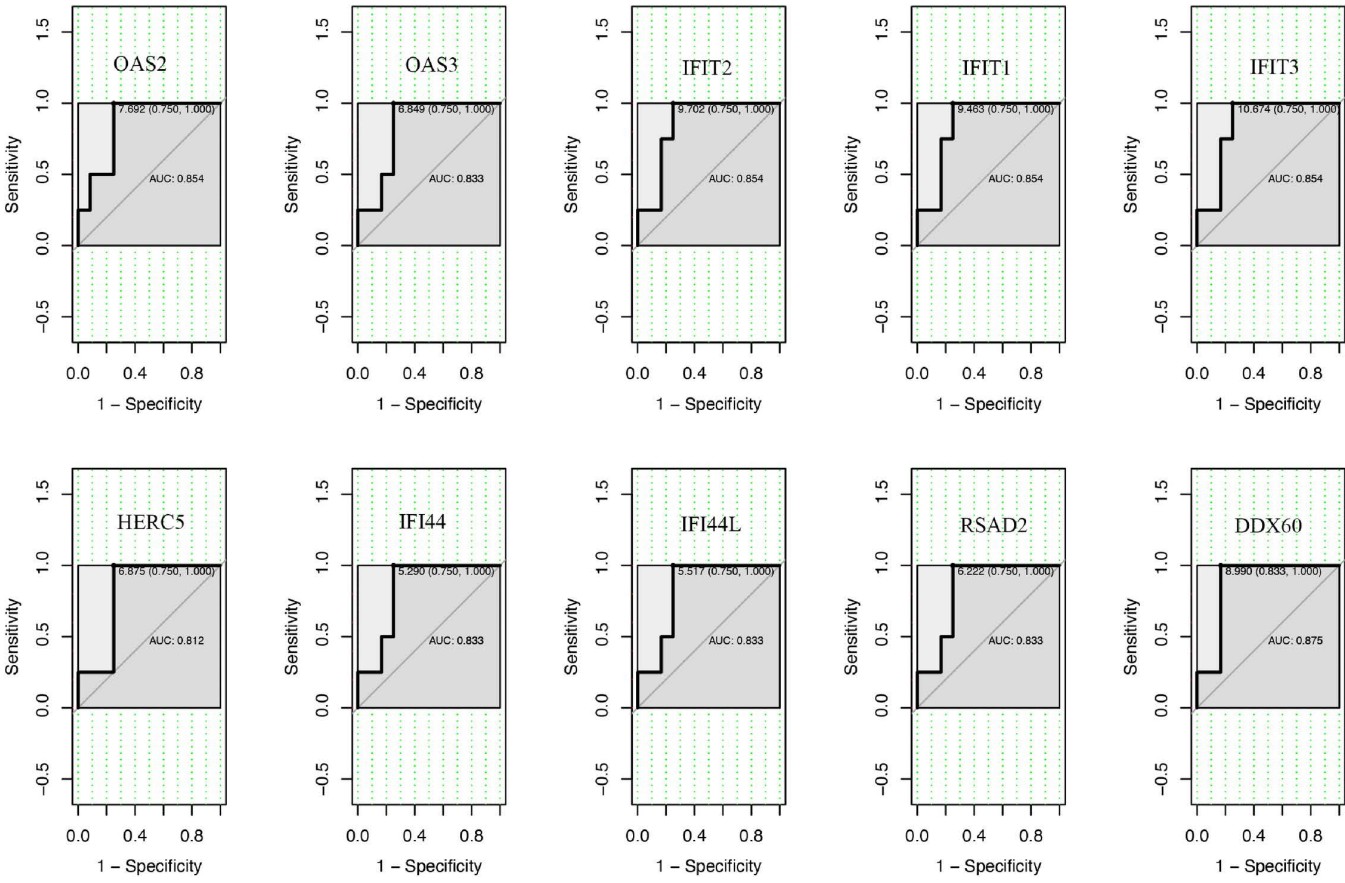

**Fig 8. ROC curves were employed to evaluate the diagnostic utility of the 10 hub genes in traumatic coagulopathy.**

Multiple hypotheses have been proposed to explain the underlying mechanisms driving the process, suggesting that tissue injury and shock work synergistically to activate the endothelium, platelets, and immune system. This activation leads to the production of various mediators that have the combined effects of reducing fibrinogen levels, impairing platelet function, and compromising thrombin generation. Consequently, these processes ultimately result in inadequate clot formation, leading to compromised hemostasis. During viral infections, similar to bacterial infections, the coagulation system undergoes activation, and in the initial stages,the activation of the coagulation cascade could potentially serve as a host defense mechanism, working to impede the spread of viruses [21]. Type I interferons (*IFNs*) play a crucial role in shaping both innate and adaptive immune responses. Activation of the Janus kinase (*JAK*)-signal transducer and activator of transcription (*STAT*) pathway through type I *IFN* signaling leads to the transcription of IFN-stimulated genes (*ISGs*) [22].

The study identified *RSAD2, IFIT1, IFIT2, IFIT3, OAS2, OAS3, IFI44*, and *IFI44L* as the eight hub genes within the *ISGs* (*IFN*-stimulated genes). *RSAD2*, also known as Radical S-adenosyl methionine domain-containing 2, is an interferon-stimulated gene that is significantly upregulated upon viral infection. It responds to both type I and type II interferon signaling through the *JAK/STAT* pathway [23]. Previous studies have demonstrated that *RSAD2* exhibits broad antiviral activity against multiple enveloped viruses. Its function as an antiviral agent has been observed in various viral infections, highlighting its potential as a promising therapeutic target for combating enveloped viruses [24]. By inhibiting the *NF-κB* pathway, the suppression of *RASD2* can effectively decrease the viability of CD19+ B cells and enhance their apoptosis. Furthermore, silencing of *RASD2* leads to a reduction in the expression of IL-10 [25].

**Table 1. Key genes and their roles in TIC.**

| Gene | Full Name | Role in TIC |
|------|-----------|-------------|
| RSAD2 | Radical S-Adenosyl Methionine Domain Containing 2 | Regulates immune response; antiviral activity. |
| IFIT1 | Interferon-Induced Protein with Tetratricopeptide Repeats 1 | Inhibits coagulation pathway activation. |
| IFIT2 | Interferon-Induced Protein with Tetratricopeptide Repeats 2 | Protects against viral infections; regulates immune pathways. |
| IFIT3 | Interferon-Induced Protein with Tetratricopeptide Repeats 3 | Regulates immune response; antiviral defense. |
| OAS2 | 2'-5'-Oligoadenylate Synthetase 2 | Activates RNase L; antiviral defense. |
| OAS3 | 2'-5'-Oligoadenylate Synthetase 3 | Facilitates viral RNA degradation. |
| IFI44 | Interferon-Induced Protein 44 | Immune mediator; contributes to antiviral response. |
| IFI44L | Interferon-Induced Protein 44-Like | Regulates immune response during viral infection. |
| DDX60 | DEAD-Box Helicase 60 | Enhances immune recognition of viral RNA; regulates biotic response pathways. |
| HERC5 | HECT And RCC1 Containing Protein 5 | Broad-spectrum antiviral activity; regulates response to biotic stimuli. |

The *IFN*-induced proteins with tetratricopeptide repeats (*IFITs*) family is one of the numerous *IFN*-stimulated gene families. Within this family, there was a cluster of duplicated loci. *IFIT1, IFIT2, IFIT3*, and *IFIT5* are present in most mammals [26]. Besides initiating a cytokine storm [27], SARS-CoV-2 infection leads to activation of the coagulation pathway by causing damage to vascular endothelial cells [28]. The presence of SARS-CoV-2 suggests potential protective effects of *IFIT1, IFIT2*, and *IFIT3* expression in gingival epithelial cells (*GECs*) against coronavirus infection [29]. Consequently, the expression of the *IFITs* family may exert an inhibitory effect on the activation of the coagulation pathway.The 2'-5'-oligoadenylate synthetases (*OAS*), including *OAS1, OAS2*, and *OAS3*, are classified as interferon-induced genes that have long been associated with an antiviral function [30]. Their downstream products can trigger the activation of RNase L, an enzyme that facilitates the breakdown of both cellular and viral components [31]. The *IFI44* gene family is recognized as a newly diversified mediator of immune responses in oysters [32].

*DDX60*, a novel DEAD-box RNA helicase, is an upstream regulator of *RIG-I* in the innate immune response. It was first discovered through microarray research focused on genes induced by measles virus infection in dendritic cells (DCs) [33]. We present experimental findings that support the co-localization of *DDX60* with the *RIG-I* protein, *RIG-I* ligand, and a stress granule marker, *G3BP*. This colocalization provides strong evidence that *DDX60* plays a role in the recognition of viral RNA by *RIG-I* [34]. In this study, we discovered that *DDX60* is involved in multiple signaling pathways related to immune regulation. Specifically, *DDX60* is implicated in the "regulation of immune effector process," "regulation of response to biotic stimulus," and "defense response to virus" signaling pathways (Fig 5a). These findings highlight the significance of *DDX60* in orchestrating immune responses against viral infection.

HECT and *RCC1*-containing protein 5 (*HERC5*) are immune proteins with potent antiviral properties. It is specifically induced in response to *IFN-α/β* signal transduction, which plays a crucial role in the innate immune response against viral infections [35]. *HERC5* demonstrates antiviral efficacy against a wide range of divergent viruses, including retroviruses such as Human Immunodeficiency Virus (HIV) and Simian Immunodeficiency Virus (SIV), as well as papillomaviruses and influenza viruses. Its ability to combat these diverse viral pathogens underscores its broad-spectrum antiviral function of *HERC5* [36,37]. In our study, we revealed the involvement of *HERC5* in key signaling pathways associated with immune regulation.

Specifically, we found that *HERC5* plays a significant role in the "regulation of response to biotic stimulus" and "defense response to virus" signaling pathways (Fig 5a). These findings underscore the importance of *HERC5* in modulating immune responses to various biotic stimuli, including viral infection.

This study has certain limitations that should be considered. First, one of the main limitations of this study was the lack of clinical data. Furthermore, despite performing a comprehensive bioinformatics analysis in the present study, we regrettably did not proceed with additional experiments. Hence, it is imperative to further investigate the specific mechanisms underlying TIC through in vivo and in vitro experiments.

## 5. Conclusions

In summary, through an extensive bioinformatics analysis, we successfully identified the DEGs.

Our study provides novel insight into the crosstalk between genes and pathways associated with TIC, laying the groundwork for future research to validate these findings in clinical setting and prospective cohorts. We identified ten hub genes and validated their diagnostic value using ROC curves. These ten hub genes are *OAS2, OAS3, IFIT2, IFIT1, IFIT3, HERC5, IFI44, IFI44L, RSAD2, DDX60.*

These eight genes are classified as Interferon-Stimulated Genes (*ISGs*), namely, *RSAD2, IFIT1, IFIT2, IFIT3, OAS2, OAS3, IFI44* and *IFI44L.*All of these genes are involved in the immune system and are relevant to antiviral defense. Consequently, they influence the coagulation system.

Our findings indicate that *HERC5* plays a crucial role in the signaling pathways related to the "regulation of response to biotic stimulus" and "defense response to viruses". *DDX60* is implicated in the "regulation of immune effector process", "regulation of response to biotic stimulus" and "defense response to virus" signaling pathways

## Author contributions

**Data curation:** Bo Li.

**Formal analysis:** Yan feng Bian.

**Visualization:** Guo Xing Lin.

**Writing – original draft:** Lingang Zhang.

**Writing – review & editing:** Jing Liu, Ying Zhou.

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
