## [Decision Letter · Decision Letter 0]

17 Jan 2025

PONE-D-24-36200Unveiling Hub Genes and Biological Pathways: A Bioinformatics Analysis of Trauma-Induced Coagulopathy (TIC)PLOS ONE

Dear Dr. 张,

Thank you for submitting your manuscript to PLOS ONE. After careful consideration, we feel that it has merit but does not fully meet PLOS ONE’s publication criteria as it currently stands. Therefore, we invite you to submit a revised version of the manuscript that addresses the points raised during the review process.

**ACADEMIC EDITOR: **

fully revise English language of the manuscriptProvide specific feedback from your evaluation of the manuscript

Our decision is justified on PLOS ONE’s publication criteria .

We look forward to receiving your revised manuscript.

Kind regards,

Monia Marchetti

Academic Editor

PLOS ONE

Journal Requirements:

2.  We note that the name of the authors are not in English. Kindly provide the full names of the authors in English, in the online submission form, as well as in the manuscript.

 “2023 Yuncheng City Basic Research Program (Free Exploration Category) Projects

Item No� YCKJ-2023052”

Additional Editor Comments:

Dear Authors

based on our review, the paper should undergo deep language revision before publication

Best regards

Academic Editor

Reviewers' comments:

Reviewer's Responses to Questions

**Comments to the Author**

1. Is the manuscript technically sound, and do the data support the conclusions?

Reviewer #1: Yes

2. Has the statistical analysis been performed appropriately and rigorously? 

Reviewer #1: Yes

3. Have the authors made all data underlying the findings in their manuscript fully available?

Reviewer #1: Yes

4. Is the manuscript presented in an intelligible fashion and written in standard English?

Reviewer #1: Yes

5. Review Comments to the Author

Reviewer #1: The paper, titled "Unveiling Hub Genes and Biological Pathways: A Bioinformatics Analysis of Trauma-Induced Coagulopathy (TIC)", explores the genetic basis of Trauma-Induced Coagulopathy (TIC), a severe condition resulting from traumatic injury. This study utilized bioinformatics techniques to analyze microarray datasets (GSE223245) from the GEO database, identifying 269 differentially expressed genes (DEGs), including 103 upregulated and 739 downregulated genes. GO and KEGG pathway analyses, along with Protein-Protein Interaction (PPI) network construction, identified 10 key hub genes closely linked to TIC. The diagnostic value of these genes was assessed using ROC analysis. The findings emphasize the potential role of these genes in TIC progression, though further experimental validation and clinical trials are necessary.

Major Concerns:

I have no major concerns with this manuscript. However, there are numerous English typos and grammatical errors that need to be addressed (see my comments in the uploaded PDF).

Minor Concerns

1. In the Introduction (page 10), the abbreviations "GO" and "KEGG" should be spelled out. This clarification should be made in the Introduction itself, not in the Functional Enrichment Analysis section.

2. The term "ROC" should be briefly explained in the Introduction.

3. Please include subheadings with appropriate numbering for the Methods section (page 11).

4. The introduction should also include any data in the literature about the role of certain known genes in the pathogenesis of TIC

5. On page 14, line 5, the sentence is unclear and appears to lack a verb. This should be revised for clarity.

6. Gene names should consistently be written in italics throughout the manuscript.

7. In the Conclusions section (page 20), you mention that your work is a groundbreaking discovery. However, given the limitations of the study (e.g., lack of clinical data and prospective cohorts), I recommend toning down this claim to provide a more balanced conclusion.

8. In Figures 3A, 3B, and 4A, the axes should be labeled.

9. The tools "CYTOHubba," "MCODE," and "CTD" should be briefly explained in the Introduction to provide context for readers.

10. There should be a table including the 10 DEGs identifies in this study and their effects of TIC

6. PLOS authors have the option to publish the peer review history of their article (what does this mean? ). If published, this will include your full peer review and any attached files.

**Do you want your identity to be public for this peer review?** For information about this choice, including consent withdrawal, please see our Privacy Policy .

Reviewer #1: No

---

## [Author Response · Author response to Decision Letter 1]

25 Jan 2025

Thank you for your detailed feedback and valuable suggestions. We have carefully addressed all the comments provided by the reviewers and the editor. The revised manuscript and a point-by-point response to the comments have been uploaded for your review.

At the same time, the author's English name was modified, and the DOI was added, sincerely appreciate your time and effort in reviewing our work and believe the revisions have improved the quality of the manuscript. Please let us know if further adjustments are needed.

DOI: dx.doi.org/10.17504/protocols.io.4r3l29bz4v1y/v1 (Private link for reviewers: https://www.protocols.io/private/77D8E398DB6C11EF88C30A58A9FEAC02 to be removed before publication.)

---

## [Editor Report · Decision Letter 1]

16 Mar 2025

Unveiling Hub Genes and Biological Pathways: A Bioinformatics Analysis of Trauma-Induced Coagulopathy (TIC)

PONE-D-24-36200R1

Dear Dr. Zhang,

We’re pleased to inform you that your manuscript has been judged scientifically suitable for publication and will be formally accepted for publication once it meets all outstanding technical requirements.

Kind regards,

Monia Marchetti

Academic Editor

PLOS ONE
---

## [Editor Report · Acceptance letter]

PONE-D-24-36200R1

PLOS ONE

Dear Dr. Zhang,

I'm pleased to inform you that your manuscript has been deemed suitable for publication in PLOS ONE. Congratulations! Your manuscript is now being handed over to our production team.

Kind regards,

on behalf of

Dr. Monia Marchetti

Academic Editor

PLOS ONE